Distribution and demographics of mysids (Crustacea: Mysida) as prey for gray whales (Eschrichtius robustus) in northwest Washington state

http://orcid.org/0000-0002-7773-0277 Allyn Elizabeth Marina liz.allyn@makah.com
Scordino Jonathan J.
Akmajian Adrianne M.
Makah Fisheries Management, Makah Tribe , Neah Bay, WA , United States of America
Venmathi Maran Balu Alagar
Electronic publication date: 2024 Jan 15
Publication date: 2024
Volume: 12
Electronic Location ID: e16587
Received 2023 Feb 7; Accepted 2023 Nov 14
Copyright: © 2024 Allyn et al.
Copyright year: 2024
Copyright holder: Allyn et al.
License: This is an open access article distributed under the terms of the Creative Commons Attribution License, which permits unrestricted use, distribution, reproduction and adaptation in any medium and for any purpose provided that it is properly attributed. For attribution, the original author(s), title, publication source (PeerJ) and either DOI or URL of the article must be cited.
License URL: https://creativecommons.org/licenses/by/4.0/

Keywords: Pacific gray whale, Foraging ecology, Predator-prey dynamics, Mysida, Eschrichtius robustus

Funding: Bureau of Indian Affairs A18AP00219 This project was funded through a Tribal Climate Resilience Grant from the Bureau of Indian Affairs (A18AP00219). The funders had no role in study design, data collection and analysis, decision to publish, or preparation of the manuscript.

==============================
Background

The movement and distribution of gray whales (Eschrichtius robustus) during the feeding season is likely dependent on the quality of foraging opportunities and the distribution of prey species. These dynamics are especially important to understand for the Pacific Coast Feeding Group (PCFG) of gray whales which spend the feeding season along the coast from northern California, USA through northern British Columbia, Canada. In Washington state, no previous work has been done to describe available gray whale prey. The main goal of this research was to initiate studies on an important gray whale prey item in northwest Washington, mysid shrimp (Mysida), by establishing a baseline understanding of mysid swarm demographics in the area and investigating patterns in gray whale and mysid presence.

Methods

Prey samples were collected during June through November 2019 and June through September 2020 using a vertically-towed plankton net at seven sites in the Strait of Juan de Fuca and seven sites in the Pacific Ocean in areas where gray whales were known to feed. Mysids collected in the samples were counted and the sex, length, species, maturity, and gravidity were documented. Patterns in gray whale and mysid co-occurrence were explored through data visualization.

Results

Seven species of mysids were observed in the survey area. In 2019, the number of mysids per tow increased steadily through the season, the most abundant species of mysids were Holmesimysis sculpta and Neomysis rayii, and sampled mysids averaged 4.7 mm in length. In 2020, mysids were abundant in tow samples in June and July but were not abundant in the remaining months of the sampling season. The average length of mysids in 2020 was 13.3 mm, and a large portion were sexually mature males and brooded females identified as H. sculpta. Throughout the survey area, the majority of whale sightings occurred later in the season in 2019 and earlier in the season in 2020, coinciding with the trends of sampled mysids.

Discussion

This study provides the first description of mysid swarm composition and temporal variation in northwest Washington. Tows were dominated by a similar assemblage of mysid species as what is observed in other areas of the PCFG range. The differences in sampled mysid assemblages between years, and the presence of whales in the survey area in times and at sites where samples with high mysid counts were collected, suggest evidence for interesting predator-prey dynamics that warrant further investigation.

Introduction

Eastern North Pacific gray whales (Eschrichtius robustus) migrate between their wintering grounds in Baja California, Mexico, and their feeding grounds in the Bering, Beaufort, and Chukchi Seas (Pike, 1962; Rice & Wolman, 1971). A subgroup of the gray whale population called the Pacific Coast Feeding Group (PCFG) does not make the full migration to the Arctic and instead shows multiyear fidelity to the coast of the Pacific Ocean from northern California, USA to northern British Columbia, Canada during the feeding season (Calambokidis et al., 2002). PCFG whales utilizing northwest Washington are of high interest to management because the Makah Tribe has proposed to resume their treaty protected right of whale hunting in the region1 .

PCFG whales have a high degree of annual site use variability throughout their range (Lagerquist et al., 2019; Calambokidis, Perez & Mahoney, 2022) and in smaller study areas within the PCFG range (Sumich, 1984; Darling, Keogh & Steeves, 1998; Dunham & Duffus, 2001; Newell & Cowles, 2006; Scordino et al., 2017). To better understand this variability, regional research groups have recently focused on identifying and characterizing the relationships between gray whales and their prey to increase our understanding of the PCFG and the role they play in the ecosystem. Studies in British Columbia and Oregon have linked gray whale behaviors and distributions on small spatial scales to changes in prey abundance, distribution, and quality (Darling, Keogh & Steeves, 1998; Newell & Cowles, 2006; Feyrer & Duffus, 2015; Burnham & Duffus, 2016, 2018; Hildebrand et al., 2022). PCFG gray whales also exhibit annual variability in body condition, which may be driven by environmental factors that affect available prey resources (Newell & Cowles, 2006; Soledade Lemos et al., 2020; Akmajian et al., 2021; Torres et al., 2022).

In northwest Washington, gray whales exhibit spatial and temporal variability in their use of coastal areas during the feeding season, and the number of whales present in the region can also vary greatly within a season and between years (Scordino et al., 2017). Better understanding of the drivers of gray whale site use in northwest Washington will help future management efforts evaluate if changes in abundance and distributions of gray whales are due to anthropogenic or natural factors. The objective of this study was to describe the assemblage of mysids observed through plankton tows, and explore patterns in mysid presence and the observed variability of gray whale site use of northwest Washington. We hypothesized that there would be similarities in when and where mysids and whales were observed within our study area because mysids are a commonly documented prey of PCFG whales (Dunham & Duffus, 2002; Scordino et al., 2017; Hildebrand, Bernard & Torres, 2021). To evaluate our hypothesis, we surveyed mysids in northwest Washington and documented species, sampled abundance, sex, size, and sexual maturity, and also surveyed gray whale distributions and abundance.

Materials and Methods

Study area

Surveys were conducted in northwest Washington State on a 6.7 m rigid-hull inflatable in the Strait of Juan de Fuca from Sekiu Point (48°16.10′N, 124°17.73′W) in the east to Cape Flattery (48°23.22′N, 124°43.70′W) in the west and in the Pacific Ocean from Cape Flattery in the north to Sea Lion Rock (47°59.58′N, 124°43.45′W) in the south as described in Scordino et al. (2017). Plankton tows were conducted at discrete sites along the established survey route (Fig. 1).

Figure 1 A map of the area where plankton tows and whale surveys were conducted along the northwest coast of Washington state during the 2019 and 2020 summer-fall gray whale (Eschrichtius robustus) feeding season.

Points represent plankton tow sampling sites. Gray lines show the area within which rates of gray whale observations per km were calculated for the Strait and Ocean regions. Map tiles by Stamen Design, under CC BY 4.0. Data by OpenStreetMap, under ODbL.

Mysid sampling

Seven sites in the Strait of Juan de Fuca and seven sites in the Pacific Ocean were surveyed to collect information about the prey base available to gray whales and the demographics of mysid swarms in this region (Fig. 1). All sample sites were close to shore in depths of less than 15.3 m and were selected based on areas where gray whales have previously been documented to feed (Scordino et al., 2017) and a desire to stratify sampling sites throughout the study region. Exact sample locations moved closer or further from the shore throughout the season to consistently sample in habitats along the edge of the expanding and contracting kelp line, as prey tows could not be conducted through the dense kelp forests. The southernmost site was adjusted between years to better reflect recent gray whale distribution patterns; in 2019 the southernmost site sampled was the South of Bodelteh Islands site and in 2020 the sampling site was moved to Ozette Island (Fig. 1). We aimed to collect one sample per site each month of the study. Sampling occurred during the gray whale feeding season during June through November in 2019 and during June through September in 2020.

Samples were collected using plankton nets similar to those used by other regional research groups (Dunham & Duffus, 2002; Burnham, 2015; Hildebrand et al., 2022). In 2019, samples were collected using a plankton net with a single 30 cm diameter hoop with a 3:1 hoop diameter to length net and 500 µm mesh. In 2020, this net was replaced with a Bongo-style net manufactured by Sea-Gear Corporation (Melbourne, FL, USA). The Bongo-style net had twin hoops measuring 30 cm in diameter with a 3:1 mouth to length ratio and a 500 µm mesh. Due to the differences in total volume sampled by these two types of net, we refrain from directly comparing sampled mysid abundance between years of the study and instead focus on month-to-month patterns and catch composition. At each site, the net was lowered over the side of the research vessel by hand until we felt the line go slack when the net hit the bottom (later confirmed through the collection of benthic species and substrate sediments), then was towed near the bottom at approximately 0.25 m s−1 through the water, accounting for current speed and direction for approximately 30 s and retrieved as fast as could be achieved by hand (approximately 0.6 m s−1). This procedure was repeated for two tows at each site, and contents of the cod end from each tow were poured into a 500 mL container comprising a single sample. If the sample did not contain mysids, or other known gray whale prey items (Nerini, 1984; Kvitek & Oliver, 1986; Duffus, 1996), then the sampling effort and absence of prey was recorded but the sample was not retained for further processing. Samples were stored in 70% ethanol before processing.

Mysid processing

For each sample, species identifications of all mysids were conducted according to Kathman et al. (1986) using a 40× dissecting microscope. Mysids were measured from the tip of the rostrum to the tip of the telson and the species, sex, maturity, and gravidity were identified. Mysids were designated as “Unknown” species if they could not be identified to species due to a damaged or absent telson. Mysids were assigned a sex based on the presence of a brood sac for females or a reproductive appendage for males (gonopod), and mysids that did not have either characteristic were designated as ‘non-brooded’. The non-brooded group therefore includes both juvenile mysids of both sexes and non-brooded adult female mysids, which were indistinguishable in this study. The characteristics of mysids in samples with high mysid counts (>1,000 mysids) were approximated by visually estimating the total count of mysids in the sample, collecting a subsample of 100 mysids from that sample, and identifying the species, sex, length, maturity, and gravidity of the subsample. The demographic composition of the subsample was then used to characterize the full sample. The samples that were characterized via subsampling were included in all subsequent analyses and can be inspected in our publicly available data repository (Allyn, Scordino & Akmajian, 2023). Other non-mysid organisms (including some species of crab larvae known to be gray whale prey items) collected during sampling were identified to the lowest taxonomic level possible using Shanks (2001) but were not included in this analysis; the non-mysid data are available in Supplement 1 and the Mendeley data repository (Allyn, Scordino & Akmajian, 2023).

Gray whale surveys

Gray whale sightings were recorded throughout the study period using the methods described in Scordino et al. (2017). In summary, the time and location were recorded at each sighting and the observed gray whales were photographed with the goal of collecting photographs for photo-identification along the dorsal flank of both sides using digital SLR cameras with 100–400 mm lenses. Photographs of gray whales were sent to Cascadia Research Collective (Olympia, WA, USA) for comparison to their catalog of individually identified gray whales from the west coast of the contiguous USA and Canada. Each whale photographed was processed according to established photo identification procedures and was either matched to an existing whale and identification number in the catalog, assigned a new identification number if no match could be made, or left unidentified if photos of the whale were of insufficient quality (Calambokidis et al., 2002; Calambokidis, Pérez & Laake, 2019). Individual identifications were used to determine the total number of unique whales observed in the survey area in a given time period, since whales can be sighted more than once during the same day or on subsequent days throughout the season. Surveys were conducted under Marine Mammal Protection Act research permits granted by NMFS to author JJS (23970 and 19430).

Data exploration

To explore potential patterns between mysid and whale presence, we performed standardization to account for different levels of effort between study months and between types of data collection (whale surveys and mysid samples). We calculated the average number of uniquely identified whales observed per km surveyed per day. We eliminated any whales that were observed engaging in behavior that was not foraging (e.g., traveling or socializing). This filtering excluded whales where we could be reasonably certain the whale was not engaging in foraging and therefore may not be responsive to the available prey reserves in the area. Whale sightings were split into two separate regions: the Strait region encompassed the area from Cape Flattery to just east of the Chito Beach mysid sampling site (48°17.74′N, 124°23.79′W), and the Ocean region encompassed the area from Cape Flattery south to approximately Sand Point (48°6.79N, 124°44.30W; Fig. 1). The average number of unique whales identified in each region was calculated per study month.

The weight of each sampled mysid was estimated using an established length-weight relationship developed in the San Francisco Bay estuary for ethanol-preserved mysids (Burdi et al., 2021). Mysid counts and biomass were calculated per sample and averaged for each region and day for each study month.

We used data visualization tools to explore patterns in this small dataset. Data analysis and visualization was conducted using Microsoft Excel and R version 4.3.1 (R Core Team, 2023). R packages dplyr (Wickham et al., 2021), psych (Revelle, 2021), and tidyr (Wickham & Girlich, 2021) were used to summarize data for analysis. Data visualization and figure development was conducted with ggplot2 (Wickham, 2016), ggmap (Kahle & Wickham, 2013), ggimage (Yu, 2023), cowplot (Wilke, 2021), gridExtra (Augie & Antonov, 2017), and ggrepel (Slowikowski, 2021). All prey sample data underlying the results presented in the study are available from Allyn, Scordino & Akmajian (2023). Those interested in whale sighting data can make a request to the Makah Fisheries Management Department of the Makah Tribe (see Data Declaration for contact information).

Results

Mysid sample composition

While we aimed to collect one sample per month, sampling effort was highly variable due to a number of factors, including lost or malfunctioning equipment, inclement weather, and the impacts of the Covid-19 pandemic on staff availability (Table 1). In the collected samples, we identified prey species from 53 taxonomic groups (Table S1), including seven species of mysids (Table S2). In 2019, 11,305 mysids were collected in 72 samples from June–November, with an average of 157 mysids collected per sample. In 2020, we collected 4,878 mysids in 78 samples from June–September, with an average of 63 mysids collected per sample. In 2019, mysids were sampled in the greatest numbers in September–November, while in 2020 there was high sampled abundance in July followed by low sampled abundance for the rest of the sampling period (Fig. 2). In both years, the samples with the greatest counts of mysids were collected in the vicinity of Sail River in the Strait of Juan de Fuca, and lower mysid counts were recorded on the Pacific Ocean side of the survey area (Fig. 3).

Table 1 The number of whale survey days and the number of plankton net samples completed in each region in each month of the study conducted in northwest Washington in summer-fall of 2019 and 2020.

Year	Region	Month	Prey samples	Whale surveys	
2019	Strait	June	5	1	
July	4	7	
August	12	6	
September	12	7	
October	7	4	
November	7	3	
Ocean	June	0	0	
July	0	2	
August	6	4	
September	12	4	
October	6	3	
November	1	1	
2020	Strait	June	0	3	
July	16	11	
August	9	4	
September	12	3	
Ocean	June	2	1	
July	21	4	
August	12	0	
September	6	0	

Figure 2 Bar chart of the average number of mysid shrimp (Mysida) collected per sample along the northwest coast of Washington state during June–November of 2019 and June–September of 2020.

Stacked bars show the species composition of the mysids sampled in each month. (*) Mysids per sample in June 2019 (0.4) and August 2020 (0.3) were low but not zero.

Figure 3 A map showing the total number of mysid shrimp (Mysida) and gray whale (Eschrichtius robustus) sightings along the northwest coast of Washington state during the summer-fall of 2019 and 2020.

Points in mysid plots (left column) represent individual samples that contained mysids, with the size of the point corresponding to the total mysid abundance. Whale icons in gray whale plots (right column) represent individual whale sightings, which indicates that one or more whale was observed. In all plots, the month of the sample or sighting is represented through the color fill of the point or icon. Inset maps show better detail of the Seal and Sail/Sail River/Bullman Beach area, which is where the majority of mysids were collected during this study. Map tiles by Stamen Design, under CC BY 4.0. Data by OpenStreetMap, under ODbL.

Most of the mysids we sampled were identified as Holmesimysis sculpta (70.3%), followed by Neomysis rayii (26.8%; Fig. 2). Mysids that could not be identified to species due to damaged or missing telsons made up 1.2% of total samples. All other species each made up less than 1% of total mysids caught, including Columbiaemysis ignota, Telacanthomysis columbiae, Hippacanthomysis platypoda, Eucopia grimaldii, and Exacanthomysis davisi. H. sculpta composed a greater proportion of the mysids caught in 2020 (99.1%) than in 2019 (58.1%).

Overall, sex could be determined for 24.2% of the mysids collected in this study. In both years, more than two-thirds of the sexed mysids were male. Of the mysids that were identified as brooded females, over half were gravid (Table 2). In every month where gravid females were present in samples (all but June and July 2019 and August 2020), at least 81% of them were identified as H. sculpta. Only seven gravid N. rayii were collected in total in September, October, and November of 2019. Mysids collected in 2019 averaged 4.7 mm in total length compared to 13.3 mm in 2020 (Fig. 4).

Table 2 Total number (percentage of total or count in parentheses) of mysid shrimp (Mysida) identified to each sex designation for mysids collected using a plankton net during surveys in Northwest Washington in the summer-fall of 2019 and 2020.

Year	Total mysids	Non-brooded (% of total mysids)	Sex identified (% of total mysids)	Male (% of [return] sex identified)	Brooded female (% of [return] sex identified)	Gravid female (% of brooded females)	
2019	11,268	10,833 (96.1%)	435 (3.9%)	223 (51.3%)	212 (48.7%)	107 (50.5%)	
2020	4,878	1,392 (28.5%)	3,486 (71.5%)	2,754 (79.0%)	732 (26.6%)	504 (68.9%)	
Total	16,146	12,225	3,921	2,977	944	611	

Figure 4 A box and whisker plot showing mysid shrimp (Mysida) length in mm for each month of sample collections along the northwest coast of Washington state during June–November of 2019 and June–September of 2020.

Mysid and gray whale co-occurrence

We conducted a total of 41 gray whale surveys during June through November of 2019 and 26 surveys during June through September of 2020 (Table 1). In both years, the peaks in whale counts occurred during months when mysids were also abundant in prey samples (Fig. 3). Most of the mysids (84.9%) collected during this study came from just one section of the Strait region that encompasses the Seal and Sail Rocks, Sail River, and Bullman Beach sites. Of the 239 whale sightings that were documented during this study, 17.6% occurred within the immediate vicinity of the Seal and Sail Rocks, Sail River, and Bullman Beach sites, even though this is a relatively small section of our full survey area. Throughout the survey area, the majority of whale sightings occurred later in the season in 2019 and earlier in the season in 2020, coinciding with times when counts of sampled mysids were also higher (Fig. 3). In the Strait, the number of unique whales observed per km was generally higher in months when mysid biomass was higher (Fig. 5). The range of mysid biomass observed in the Ocean was too constrained (consisted of zero and near-zero counts) to explore patterns with whale presence.

Figure 5 Mysid shrimp (Mysida) biomass and gray whale (Eschrichtius robustus) observations from surveys conducted along the northwest coast of Washington state during the summer-fall of 2019 and 2020.

Average mysid biomass is expressed in grams, and whales are represented as the average number of uniquely identified gray whales observed per km for each month.

Discussion

This study provides the first description of mysid presence and demographics in northwest Washington. Samples were dominated by H. sculpta and N. rayii, which also dominate the mysid assemblages in nearby gray whale feeding areas in British Columbia and Oregon (Darling, Keogh & Steeves, 1998; Newell & Cowles, 2006; Feyrer, 2010; Burnham, 2015; Hildebrand, Bernard & Torres, 2021; Hildebrand et al., 2022). H. platypoda, which was first documented in British Columbia in 2017 (Burnham, Meland & Duffus, 2017), was also observed in our study area, confirming its presence in the region.

The demographic composition of mysids in prey samples differed greatly between 2019 and 2020. In 2019, H. sculpta and N. rayii were both dominant species in all months, but in 2020 the samples were almost entirely composed of H. sculpta. This difference in species composition between years may be linked to reproductive strategies and success. H. sculpta were the only mysid species observed to reproduce in the winter months in Clayoquot Sound, British Columbia (Burnham, 2015). If the high densities of mysids in fall 2019 and July 2020 in our study were met with intense foraging pressure from gray whales or other predators, N. rayii and other species of mysid may not have had an opportunity to reproduce successfully during this intense predation, while H. sculpta may have been able to reproduce over the winter months, similar to what was seen in Clayoquot Sound. While we did not explore a statistical relationship between sampled mysid abundance and whale observations due to unequal sampling between regions, months, and years (Table 1), the temporal and spatial co-occurrence of whales and mysids visible in our data (Fig. 3) suggests an interesting predator-prey dynamic that warrants further investigation into the role of gray whales in the ecosystem.

The variability in whale and mysid presence in our study area between years may be related to variability in gray whale foraging behaviors. Four distinct foraging strategies have been described in the PCFG range: benthic foraging on amphipods, epibenthic foraging on mysid swarms, pelagic foraging on crab larvae (primarily Porcellanidae), and shallow water foraging for ghost shrimp (Callianassa californiensis) (Nerini, 1984; Kvitek & Oliver, 1986; Duffus, 1996). It is likely that these different foraging strategies help explain the way gray whales distribute throughout their range, and the variability in site use that is observed in individual areas of that range. In our research area, whales could feasibly forage on multiple prey items due to the close proximity of multiple habitat types, such as the rocky bottom and kelp forests observed at Seal and Sail Rocks and Bullman Beach that are adjacent to sand and mud bottom. During our study, gray whales were occasionally observed near sample sites creating mud plumes, which are usually associated with benthic foraging on sediment-dwelling amphipods (Nerini, 1984). We also collected large numbers of Porcellanidae larvae, a known gray whale prey item, in the Ocean region during October 2019 and August 2020. The presence of these additional feeding strategies and prey types in our study area could explain the presence of feeding whales in areas of low counts of sampled mysids, as they could be targeting a completely different assemblage of prey species.

The discrete nature of our sampling methods and the patchiness of mysid swarms are likely also responsible for some of the difficulty documenting a direct relationship between whales and their mysid prey through our data exploration efforts (Clutter, 1967). When looking at the presence of whales during sample collection, there were 10 samples with high mysid counts that were accompanied by no whale presence, and 21 samples with zero mysids where whales were present. In instances when whales were present and appeared to be foraging, but mysids were not collected, it is possible that mysids were present in the general vicinity of our sample sites but were not collected during sampling. Instances when we collected a sample with mysids but did not observe whales feeding in the area could be caused by several factors, including the existence of adequate prey elsewhere in their feeding range. Hildebrand et al. (2022) posit that widespread low abundances of N. rayii and other high-calorie prey items throughout the PCFG range could explain patchiness in gray whale foraging behaviors (Feyrer & Duffus, 2015; Hildebrand et al., 2022). This complements the idea developed by Darling, Keogh & Steeves (1998) that while gray whales collectively use the entirety of their foraging range and multiple prey types within a feeding season, foraging pressure at any given time and place can appear patchy.

Future analyses of prey sample data in this region would benefit from additional data collection such as simultaneous gray whale focal follow effort and scat sampling to directly study diet, using drone-based monitoring in order to document behavior and enable researchers to differentiate between foraging strategies, and improved methods for sampling and quantifying mysids and other prey items perhaps using drop cameras or sonar. This additional information would allow us to better understand how whales use this area, which prey resources they rely on most, and what this means for the status of the PCFG.

Conclusions

This study improves our understanding of gray whale prey in the PCFG range by providing the first information on the demographic composition of sampled mysids, spatial and temporal variation in sampled mysid assemblages, and patterns of concurrent whale distribution in northwest Washington. The different mysid assemblages observed in 2019 and 2020 highlight the variation of prey in the PCFG range, and the complex dynamics that likely determine gray whale foraging distribution within this range. Our results indicate that additional prey sampling and more detailed gray whale behavioral observations could help elucidate this predator-prey relationship. With this study we establish a baseline understanding of mysid swarm composition in northwest Washington and encourage future research on gray whale foraging behavior throughout the PCFG range to better characterize their role in the ecosystem.

Supplemental Information

Supplemental Information 1 A full list of the taxonomic groups caught during plankton net sampling along the coast of northwest Washington in the summer-fall of 2019 and 2020.

Total is the sum of all observations over both years. 2019 Per Sample and 2020 Per Sample show the average number of that taxonomic group collected per sample during each year. “Unk” is an abbreviation of “unknown” and is used to indicate that the organism was unable to be identified lower than the presented designation due to damaged identification features or lack of access to specialized expertise.

Click here for additional data file.

Supplemental Information 2 The average number of each species of mysid shrimp (Mysida) caught per sample for each month of plankton net sampling in northwest Washington state during summer-fall 2019 and 2020.

Unknown mysids were unable to be identified to species due to damaged or missing telsons.

Click here for additional data file.

The authors would like to acknowledge Dr. Rianna Burnham and Dr. David Duffus for providing training in mysid identification and prey sample processing. Thanks also to the 2019 Makah Fisheries Interns, Angelina Woods, Laney Keyes, Tobias Croy, and Kaeden Butterfield, for assisting with sample collection in 2019 and to Charlotte Shaw for sample collection assistance in 2020.

Additional Information and Declarations

Competing Interests

Author Contributions

Animal Ethics

Field Study Permissions

Data Availability

1 More information available at https://www.fisheries.noaa.gov/west-coast/marine-mammal-protection/makah-tribal-whale-hunt.

The authors are all employees of the Makah Fisheries Management department of the Makah Tribe.

Elizabeth Marina Allyn conceived and designed the experiments, performed the experiments, analyzed the data, prepared figures and/or tables, authored or reviewed drafts of the article, and approved the final draft.

Jonathan J. Scordino conceived and designed the experiments, performed the experiments, authored or reviewed drafts of the article, and approved the final draft.

Adrianne M. Akmajian conceived and designed the experiments, authored or reviewed drafts of the article, and approved the final draft.

The following information was supplied relating to ethical approvals (i.e., approving body and any reference numbers):

The Makah Tribe does not have an Institutional Animal Care and Use Committee or require approval through another similar body. The National Marine Fisheries Service reviewed and approved our research methodologies for gray whale surveys and granted Marine Mammal Protection Act research permits 23970 and 19430.

The following information was supplied relating to field study approvals (i.e., approving body and any reference numbers):

The National Marine Fisheries Service reviewed and approved our research methodologies for gray whale surveys and granted Marine Mammal Protection Act research permits 23970 and 19430 to author Jonathan J. Scordino.

The following information was supplied regarding data availability:

The data from plankton tows conducted during this study are available at Mendeley Data: Allyn, Liz; Scordino, Jonathan; Akmajian, Adrianne (2023), “Plankton tow data from nearshore coastal Washington State 2019-2020”, Mendeley Data, V1, DOI 10.17632/xpsgkxdnwk.1.

https://data.mendeley.com/datasets/xpsgkxdnwk/1.

The whale sighting data is not publicly available as it is sensitive information restricted to the Makah Tribe. It can be requested from the Makah Fisheries Management Department of the Makah Tribe, 150 Resort Drive, Neah Bay, WA, USA, or via email: info@makah.com.

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
