# Peer review of "Distribution and demographics of mysids (Crustacea: Mysida) as prey for gray whales (Eschrichtius robustus) in northwest Washington state"

_PeerJ, doi:10.7717/peerj.16587_

## Round 0.1 · original submission · Major Revisions

The most important issue in this manuscript is the lack of environmental data as recommended by a reviewer, it can give a significant impact on the distribution of mysid and other prey organisms. Relational analysis between the distribution of mysid and environmental data directly above the seabed is essential for considering their occurrence and distribution.

The manuscript has been recommended for revision by 3 reviewers, they have given some valid comments for your manuscript to improvise. Please follow all reviewers' comments carefully and address all the issues in your revision. Based on all the reviewers' comments, I advise major revisions.

Reviewer 1 ·

Basic reporting

The manuscript is clearly written and easy to follow. There are some literature that the authors might want to consult/reference including:
Mauchline 1980, Burnham and Duffus 2022, Feyrer and Duffus 2011, 2014,

The raw data was not shared, but some was referenced to a Mendeley library

The results and presentation are self-contained, although there are some comments below that address the clarity of the hypothesis

Clearer statement of hypothesis – is this a prey paper, a mysid as prey paper or a mysid paper

Experimental design

This manuscript meets the aims and scope of the journal, and has defensible, robust and repeatable methods. The clarity and impact of the findings could be highlighted given some editing of the paper.

It would be helpful for the authors to more clearly state the research question – as there are a number of things being addressed by the work conducted, any perspective and interpretation would be a relevant and worthy addition to the gray whale literature.

The methods were rigorous, but could be better described.

Validity of the findings

Greater clarity to the research aim and hypothesis could help show how the authors intended to add to the existing body of literature on gray whales and the PCFG.

Not all underlying data has been presented – and there are likely some analyses of the data that was collected that could add to the results presented to strengthen the conclusions.

Conclusions are sound, however having a more clearly stated hypothesis would better guide both the introduction and the conclusion of the piece.

Additional comments

Be careful with the use of phrases such as prey density and prey availability. In this piece you are really just presenting the number of mysids per tow – which can be influenced by many things. One may be the density of the swarm, but it also could be mysid size, species, startle response, how the swarm is sampled (i.e. through the middles, compared to along an edge). Also prey availability should take into account the abundance and swarm density but also the size of the individuals

There are some places were what is written in one section seems to be contradicted in another section – be careful to check this.

Generally more detail for the methods are needed, and greater justification of the methods and definitions (e.g. the high and low density of swarms) is also needed.

Background, Line 23: foraging habitat use patterns
Methods, Line 26: were these vertically-towed or horizontally-towed plankton net samples
Results, line 31 (and throughout): Be careful with comparators such as ‘small’ – small compared to?
Discussion: The discussion here does not quite match the discussion in the main piece, and this speaks more to the results.

Introduction:

General: Introduction doesn’t flow – a thesis behind the work is not clear – is it the use of the area, the thresholds of use of the area, or some examination of prey and foraging by gray whales in the context of the UME. I think if this is a predator-prey paper the introduction should be focused there, and perhaps things like body condition and the UME could be part of the discussion and rationale and importance of the study and the reasoning behind understanding both the prey reserves but also the relationship between predator and prey.

Lines 65-67 seemingly contradict Lines 45-48.
Line 85-88: I am not sure it is species composition alone – it is the species composition that influences the swarm dynamics, but be careful not to make it sound that whales are selecting based on the species within the swarm (again in Line 265-266).

Methods:

There are some results presented in the methods, e.g. lines 104-106, 132.
Generally the inclusion of species other than mysids is a little confused – either include a little more detail about the species, proportion of the samples etc. (making the paper perhaps more a general prey paper) or maybe just make mention there were other invertebrates caught in the net including mysids, and give your focus to that analysis.

Lines 105-106, 109-111 – why was this method of keeping to the edge of the kelp taken rather than keeping the sample sites consistent? Why did the monthly sampling of sites vary?
Lines 120-121: Why 2 replications per site. What is the 500 mL container was filled in the first sample?
Line 132: Why were large samples sub-sampled? Was Mysid number per sample was counted – but not all mysids were speciated etc.? or was the number of individuals estimated for large samples, and so the decision made to subsample?
Line 135-136: What does this mean?
Line 143: How was this specifically done so that it was comparable each time. More detail in this section would be helpful
Line 143-149: If photo-identification is going to be included a bit more of the methods should be included here.
Line 153: How is overall mysid density of the study area defined. Do you think this relationship is occurs on a finer scale as well as on the scale of the study area?
Line 155: Why within 30 mins?


Results
Line 168-182: Catch composition – with proportion of mysid-non mysid and proportions of species, body sized individuals, and gravid females would be more comprehensible than the numbers, and allow easier comparison.
Line 193: Do you mean seasonal, or monthly? You essentially sampled in the summer – and go on to describe differences in the monthly data

Lines 197- 199: A comment – In 2019 you collected mostly small bodied individuals that are not a source or not a good source of food for gray whales. I would suggest that they reason that the whale numbers were highest in the latter part of the summer is because of the growth of the mysids through the summer allowing them to become a more viable prey item – and that the mysids that you see in 2020 are the matured version of the 2019 individuals. What would be interesting would be for you to comment on the predation prior to these sample years. My guess would be that predation was heavy in 2018 if not several years prior given what has been seen for gray whales and mysids in other foraging locations.
It seems that the largest samples were collected after the swarms had stabilised and individuals in the swarms had reached some form of maturity. Also later in the summer you may be looking at a prey resource that has seen a spring/summer brood that is maturing.

There is no mention in the paper of the limitation of body size as prey – in fact you state that whale were observed foraging in proximity to swarms regardless of size (Line 247-248, which contradicts Lines 239-241). This is not as prevalent for prey species such as crab larvae that conglomerate together, but for mysids that also have a startle/avoidance response the body size could be influential to the foraging behaviours that are noted

Line 208-210: Transect or whale data to show the average number of whales per survey, their use, and residency time in the area would add to the paper, and this will allow you to comment on the potential prey reserves and the predator-prey relationship more clearly. If these details are not include suggest removing the photo-identification section in the methods, unless it is used to help confirm whale number per survey, for example.
Were whale surveys and mysid collection always concomitant – or where there more surveys in between the monthly prey sampling? The use of chi-squared shows that behaviour is not random (as would be suggested from previous research), but were there any correlations between say the average body size or proportion of certain species or proportion of the sample that was mysid or non-mysid with whale presence?

Discussion:
Line 239: Swarm density is impacted by body size – mysids swarm based on body size so it is expected that the smaller the body size the greater the number of individuals that may be caught. Smaller individuals may also be more limited in their ability to swim rapidly to escape capture – however these smaller bodied individuals do not really represent a viable food source

Lines 275- 277 – but were they observed or noted to be foraging and throwing up mud plumes in close proximity. Generally mysid habitat and amphipod habitat are pretty distinct.

Figures:

Figure 2 – not completely clear – is there a better way to present this – perhaps even in a table for each month

Figure 3 – may be better to represent this proportionally rather than absolute numbers
Same comment for figure 4 – or a box and whisker plot maybe

What is the correlation value between the variables in Figure 5 – that might be interesting to state
Where did the density definition come from?

Reviewer 2 ·

Basic reporting

This paper is a valuable record of simultaneous surveys of the distribution of mysids, the main prey of grey whales, and the appearance of grey whales, aimed at understanding their feeding patterns. The basic research design is relatively simple, and the results and interpretations are presented in clear English with figures and tables. However, some parts of the discussion supporting the conclusions are inadequate and need to be improved in the following ways for publication.

Experimental design

Mysids tend to form swarms/schools and distribute near the seabed, especially during the daytime, making quantitative collection difficult. To collect them, researchers generally use a benthic sledge net that can be towed along the seabed. In contrast, the authors adopted a method of towing a standard plankton net (Bongo net) in the near-bottom layer. However, it is unclear whether the net used by the authors can collect samples directly above the seabed, and some kind of verification is needed. For example, the presence or absence of benthos living in close proximity to the seabed or the degree of sediment contamination would be good indicators for determining the sampling layer. In addition, to enable comparisons with other studies, the density of mysids should be expressed per unit of water volume rather than per sample. This is also important for establishing criteria for suitable feeding environments for whales in the future. Furthermore, the total number of individuals collected in each month shown in Table S2 is meaningless as the sampling frequency of each month differs between 2019 and 2020. Here appropriate standardization of the data is necessary. Also, detailed explanations are required in the main text for the extrapolated or estimated values. Additionally, the lack of environmental data such as water temperature and salinity near the bottom, which have a significant impact on the distribution of mysids and other prey organisms, is a major problem. Relational analysis between the distribution of mysids and environmental data directly above the seabed is essential for considering their occurrence and distribution, and improvements are needed in this regard.

Validity of the findings

I largely agree with the findings that the distribution of grey whales matched the abundance of mysids is valuable. However, the possibility of top-down effects of whale feeding on the distribution of mysids needs to be carefully examined. First, mysids have strong environmental preferences and are known to exhibit a prominent zonal distribution along environmental gradients such as the types of substrates, the properties and currents of water masses that change in the on-offshore direction, as well as, seasonally (Clutter 1967, Ecology, 48: 200-208). The significant changes in the species composition and density of mysids shown in this paper could be therefore caused by the inter-annual or seasonal variation of this zonal distribution pattern. Ideally, it is desirable to compare the seasonal distribution changes at multiple points with different depths, but if it is difficult, a comparison with environmental factors would be effective. At the very least, the authors should examine whether the water temperature, salinity, and the degree of kelp coverage at the sampling sites in 2019 and 2020 differed significantly, and if there are differences, discuss the possibility that the annual differences in the use of mysid habitats could have affected the sampling results. If this is the case, the discussion in this paper would become even more interesting, as it would suggest that environmental factors that affect the distribution of mysids also have a significant impact on the habitat use for feeding in grey whales.

Additional comments

Table S1 and Table S2 appear to be cited in reverse in the text, so a correction is needed. Additionally, some of the species names of the planktonic crustaceans listed in Table S2 have only the abbreviated genus name mentioned, except for some species. It would be helpful to spell out the full scientific name of these species in the tables.

Reviewer 3 ·

Basic reporting

This is a good piece of work, and it provides further informations on the demographic composition of mysid assemblages, spatial and temporal variation in prey availability, and patterns of whale distribution in northwest Washington. The foraging ecology data of marine mammals are very crucial information in view of conservation of highly vulnerable animals. Even though the MS has some issues, especially in the methodology, introduction and discussion (all highlighted in the PDF file), this paper should be published after a MAJOR REVISION.

Experimental design

Issues are highlighted in the attached PDF

Validity of the findings

Accaptable

Annotated reviews are not available for download in order to protect the identity of reviewers who chose to remain anonymous.

---

## Round 0.2 · Minor Revisions

You have well revised and improvised the manuscript, however please follow the reviewers comments for their minor revision suggestions. Figures and Tables need to be improved to make them more comprehensive. One reviewer commented that figures 2 and 4, data from different years should be shown in different panels and the current version is confusing. I suggest authors to follow all their comments carefully before your resubmission.

Reviewer 1 ·

Basic reporting

Line 75, 172, generally throughout: I would suggest that the authors consider throughout when they use the words abundance and if this is what they mean, or whether they are testing presence. Future surveys with echosounders, for example, may better help determine presence (and not missed swarms when sampling) and abundance of mysids. Here you are really talking, I believe, about relative abundance in the samples that have been taken- with this presumed to be representative of the swarm as a whole

Lines 97-98: Read more like results. The intent was to sample monthly, but due to a number of factors that was not possible

Line 104-105: I don’t think this detail about the net is necessary – sufficient to say that the net was changed between years, but it should not impact results if no absolutes are given (just have things reported as proportions)

Line 112: knot should be written out in full and then abbreviated.
Try and keep measures consistent e.g. kts on line 112 and then m/s on line 114

Line 113: I think sufficient to just say ‘approximately 30 seconds’ and not 30-count as well

Experimental design

Line 92: Why was the sample location chosen to move along the kelp line (and so presumably more coastal during the fall/winter as the kelp dies away) and not maintain consistent sampling sites.

Line 111-112: This makes it sound as if the net was dragged along the bottom. If the net hit/was resting on the bottom slack in the line could also perhaps be used to indicate this – but this also seems like an unneeded detail.

Line 117: is there a distinction between known and potential prey items

Line 127: does non-brooded exclude juveniles that may be male but with a less obvious appendage present?

Line 129-130: Was the ability to estimate the number of mysids tested or verified in any way? Why was 1000 chosen (fine to be arbitrary)

Line 141-156: I think it likely matters less here that the whales are unique individuals than purely the number of foraging whales at any one time, to estimate foraging pressure/removal from gray whales on mysids – or conversely the potential prey reserve available to sustain a certain number of whales.

Line 162: This leaves some ambiguity on the behavioural state – is the behavioural state here assigned based on the location, for example. I think or clarity just say where the whales were known to be foraging/feeding

Line 170-173, Lines 264-266: I appreciate the biomass work – but again care should be taken to not make assertions or conclusions that can not be supported given that the work is does not absolutely confirm the presence/size of swarms due to the nature of net sampling

Line 198: Should this be less than 2% to sum to 100 overall?

Line 201: Any conclusions based on gender should be considered with care give that less than a quarter of the samples were able to be conclusively assigned

Line 219: How did you distinguish in the biomass between lots of little mysids compared to a few larger bodied – these may have similar values but do not represent the same level of food source for whales

Line 223: Mysid presence

Validity of the findings

No comment

Additional comments

The authors have put in great efforts to address the comments made in the first round of reviews. The methods and processes are clearer, and there is less ambiguity as to the thesis of the paper. There are, however, still some minor edits/changes that could be made to this second round draft

Reviewer 2 ·

Basic reporting

The revised manuscript is clearly written and easy to follow. Figures and Tables could be improved to make them more comprehensive.

Figures 2 and 4: data from different years should be shown in the different panels. The current version is very confusing since it looks like successive variations.

Table 1: Need amendment since it is less informative and does not match with the statement in the text (L189-192 etc,)

Experimental design

Although the sampling design was not perfect to test the hypothesis, the authors discussed well about its shortage and possible errors, which will improve and enhance further research program.

I guess adequate estimation of mysid abundance will be key to assessing the feeding ground of grey whales in the future research program. In this context, the author may be able to add suggestions to achieve this such as night-time sampling (mysids generally shows nocturnal emergence to the water column), monitoring with video camera, e-DNA, etc.

Validity of the findings

The authors revised the manuscript extensively in order to accommodate the comments raised by the referees. As admitted by the authors, the sampling design and data set were not perfect, I consider it important to show the variability of potential prey abundance and composition with grey whale occurrence.

Additional comments

I felt the statement in the background "The movement and distribution of grey whales (Eschrichtius robustus) during the feeding season is largely determined by the quality of foraging opportunities and the distribution of prey species." is a bit contradictory since this is a central question of the paper.

Reviewer 3 ·

Basic reporting

The authors have adequately addressed the reviewer’s comments and incorporated the suggestions. Now it is in an acceptable form. Only one small suggestion on the title (Please delete the word 'item') (please see the PDF).
Thank you
Reviewer

Experimental design

Acceptable

Validity of the findings

Acceptable

Additional comments

The authors have adequately addressed the reviewer’s comments and incorporated the suggestions. Now it is in an acceptable form. Only one small suggestion on the title (Please delete the word 'item') (please see the PDF).
Thank you
Reviewer

Annotated reviews are not available for download in order to protect the identity of reviewers who chose to remain anonymous.

---

## Round 0.3 · accepted · Accept

Authors have thoroughly revised the manuscript and hence it can be accepted for publication.